# Key-Grid: Unsupervised 3D Keypoints Detection using Grid Heatmap Features

**Chengkai Hou**[1,3]**, Zhengrong Xue**[1,2]**, Bingyang Zhou**[4]**, Jinghan Ke**[5]**,
Lin Shao**[6]**, Huazhe Xu**[1,2,7]

[1]Shanghai Qizhi Institute    [2]Tsinghua University    [3]Peking University
[4] The University of Hong Kong    [5] University of Science and Technology of China
[6] National Unversity of Singapore    [7] Shanghai AI Lab

## Abstract

Detecting 3D keypoints with semantic consistency is widely used in many scenarios such as pose estimation, shape registration and robotics. Currently, most unsupervised 3D keypoint detection methods focus on rigid-body objects. However, when faced with deformable objects, the keypoints they identify do not preserve semantic consistency well. In this paper, we introduce an innovative unsupervised keypoint detector **Key-Grid** for both rigid-body and deformable objects, which is an autoencoder framework. The encoder predicts keypoints and the decoder utilizes the generated keypoints to reconstruct the objects. Unlike previous work, we leverage the identified keypoint information to form a 3D grid feature heatmap called **grid heatmap**, which is used in the decoder section. Grid heatmap is a novel concept that represents the latent variables for grid points sampled uniformly in the 3D cubic space, where these variables are the shortest distance between the grid points and the "skeleton" connected by keypoint pairs. Meanwhile, we incorporate the information from each layer of the encoder into the decoder model. We conduct an extensive evaluation of Key-Grid on a list of benchmark datasets. Key-Grid achieves the state-of-the-art performance on the semantic consistency and position accuracy of keypoints. Moreover, we demonstrate the robustness of Key-Grid to noise and downsampling. In addition, we achieve SE-(3) invariance of keypoints though generalizing Key-Grid to a SE(3)-invariant backbone.

## 1 Introduction

Representing objects through a set of 3D keypoints is one of the most popular and intuitive approaches to compress and comprehend 3D objects [31; 26]. Effectively exhibiting their utility, 3D keypoints have contributed to the success of a number of downstream tasks, including pose estimation, shape registration, object tracking in computer vision [15; 27; 4; 8; 44; 28; 34], as well as various kinds of robotic manipulation tasks [40; 1; 16].

To make the detected keypoints as capable and accessible as possible, the research community is now concentrating on the unsupervised learning of semantically consistent keypoints on 3D point clouds. The implication of *semantic consistency* is typically twofold: the keypoints should be located at the semantically salient parts of the objects; they are also desired to be aligned within the same category even under large shape variations among diverse 3D object instances. To achieve these objectives, previous works [26; 9; 41; 3; 46] often adopt autoencoder frameworks to facilitate self-supervised training, where the encoders serving as the keypoint predictor are backbone networks [21; 22] generalizable to the shape variation, and the decoders reconstruct the input shape conditioned on the predicted keypoints. Since neural networks are in general better at compression than generation [20],

---

[0]Corresponding to: `Houchengkai1998@outlook.com`, `huazhe_xu@mail.tsinghua.edu.cn`

38th Conference on Neural Information Processing Systems (NeurIPS 2024).

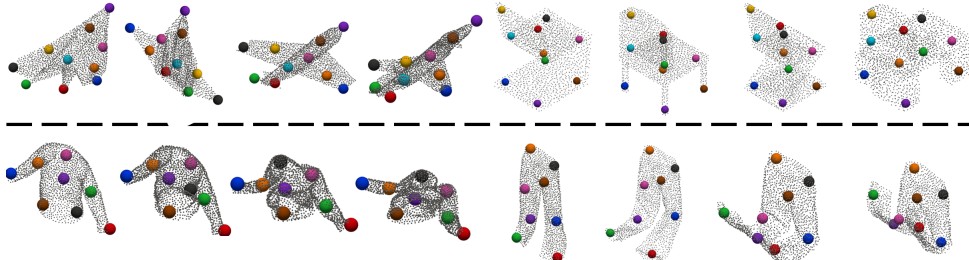

Figure 1: **Examples of the keypoints detected by Key-Grid.** The detected keypoints preserve semantic consistency under: (Top) large intra-category shape variations of rigid-body objects from the ShapeNetCoreV2 [2] dataset; (Bottom) dramatic deformations of soft-body objects from the ClothesNet [49] dataset.

the primary challenge of this pipeline lies in reconstructing the entire point cloud from a few estimated keypoints. Thus, the state-of-the-art (SOTA) detectors [26; 9] lay emphasis on leveraging different priors on 3D structures (e.g., "skeleton" in Skeleton Merger [26], and "cage" in KeypointDeformer [9]) so that the 3D object shapes can be more reasonably approximated by the information from the detected keypoints alone.

While maintaining semantic consistency is already demanding under the shape variations of rigid-body objects (e.g., those in the ShapeNetCoreV2 [2] dataset), it would be even more challenging if deformable objects are also taken into consideration. For instance, when detecting keypoints on the trousers, if one detected keypoint is located at one of the trouser legs, then the keypoint is desired to follow the motion of the trouser leg in the process of the trousers being folded (shown in Figure 1). Note that the shape variation caused by deformation is so dramatic that even the outline of the object shape has been completely altered, indicating a shift in the spatial and geometric structure of the keypoints. Despite the difficulty, with a growing interest in deformable object manipulation [1; 24; 25] in robotics as well as the emergence of large-scale datasets for deformable objects [49; 6] in computer vision, it is of increasing significance to develop a keypoint detector that is equally effectively when faced with deformable objects.

In this work, we present **Key-Grid**, an unsupervised keypoint detector on 3D point clouds aiming for semantic consistency under the shape variations of both rigid-body and deformable objects. In accordance with the prevailing practice, Key-Grid uses an autoencoder framework. In response to the potentially shifted geometric structure of the keypoints brought by deformations, we propose to embed the information of the predicted keypoints into a dense 3D feature heatmap. To be more specific, we first uniformly sample a large number of grid points in the shape of a 3D array. Then, we assign a feature to each grid by calculating the shortest distance from the grid point to the connected lines of all the keypoint pairs (i.e., the "skeleton" [26] of the keypoints) and multiplying it by the weight of the connected lines. Finally, when the decoder attempts to reconstruct the point cloud, coarse-to-fine features are extracted from the dense grid feature heatmap, where the extracted point coordinates are in line with the hierarchical point sets in the PointNet++ [22] modules of the encoder. Intuitively, the grid heatmap can be viewed as a dense extension of the "skeleton" where its undefined blank spaces are smoothly extrapolated. Functionally, the grid heatmap constitutes a continuous feature landscape across the entire 3D space, providing richer and more stable geometric descriptions of the object. This could be vitally beneficial when the object undergoes intense shape variations, such as cloth deformations.

Empirically, the experimental results show that Key-Grid not only achieves SOTA performance for rigid-body objects in the popular ShapeNetCoreV2 [2] dataset but also surpasses the previous SOTA [9] by $8.0\%$ and $9.1\%$ for objects with dropping and dragging deformation respectively in the recently proposed ClothesNet [49] dataset. Meanwhile, Key-Grid is found to be robust to noise or downsampling operations. Additionally, we also show that Key-Grid can be easily extended to an SE(3)-equivariant version when it is integrated with the USEEK [40] framework. We are committed to releasing the code.

## 2    Related works

**Deformable object datasets.**    While there is an increasing number of large-scale 3D dataset repositories such as ShapeNetCoreV2 [2], PARTNET [18], SAPIEN [37], and Thingi10K [50], only

a few datasets contain deformations from the same model. Among them, the deformations from the same piece of clothing are diverse and have practical research significance. For instance, Deep Fashion3D [6] contains around 2,000 3D models reconstructed from 563 real garment instances in different poses. A subset of ClothesNet [49] contains around 3,000 3D garment meshes which can be directly loaded into differentiable simulations such as DiffclothAI [45] to generate various deformations after operating like dropping, folding, or dragging.

**Unsupervised keypoint detection.** There have been various approaches proposed for supervised estimation of 3D keypoints using manually annotated keypoints [35; 5; 14; 52; 13; 42]. In contrast to supervised methods, our approach is unsupervised, meaning it does not rely on manually annotated keypoints. Thus, we review methods that adopt an unsupervised approach for estimating 3D keypoints. USIP [12] is the first detector that identifies 3D keypoints in an unsupervised manner by minimizing the chamfer distance between detected keypoints in randomly transformed object pairs. Canonical Capsules [29] is a similar approach that feeds pairs of a randomly translated copies of the same object into a network to detect keypoints. Following USIP, SC3K [51] also uses a random rotation to create two transformed visions of objects and generate the corresponding keypoints by mapping the keypoints of each version back to the original object. Another way to estimate 3D keypoints is proposed by Chen et al. [3]. They encode the point cloud as a set of local feature and input it into a novel structure model to generate the possibility of keypoints. Recently, Fernandez et al. [11] proposed a novel method that distinguishes between *Node branches* and *Pose and coefficient branches* to find the optimal keypoints of an object.

However, these methods do not consider the geometric structure information that keypoints can represent. When they encounter irregular shape of objects, such as airplanes and cars, the keypoints they identify will lose the crucial information about the object. Skeleton Merger [26] generates the skeletons of an object connected by keypoints and uses the Composite Chamfer Distance (CCD) to make these skeletons close to the original point cloud. This makes the detected keypoints to represent the important structural information of the object. Yuan et al. [46] propose a similar way that generates keypoints by utilizing skeletons from two objects within the same category to reconstruct mutually. USEEK [40] utilizes a teacher-student network, where the teacher module is based on Skeleton Merger and the student module employs a SE(3)-invariant backbone network, SPRIN [43]. LAKe-Net [32] uses detected keypoints to achieve the shape completion by locating them to generate a surface skeleton and refining the shape of the object. KeypointNet [30] learns category-specific 3D keypoints using depth and 2D position information from a pair of 2D images. For keypoint detection on deformable objects, KeypointDeformer [9] aligns the shape of the source object to the target object by utilizing the difference in keypoints positions between them and propose a novel loss function that encourages keypoints to distribute well and keep semantic consistency.

## 3 Method

In this section, we propose Key-Grid, an unsupervised keypoint detector on 3D point clouds based on the autoencoder architecture. Figure 2 shows the overview of Key-Grid. In the following section, we provide a detailed explanation of the key ingredients in the Key-Grid: an encoder that predicts keypoint locations in an input point cloud; a grid heatmap is a 3D feature map used to capture the geometric structure of deformable objects by computing the shortest distance from points uniformly sampled in 3D cubic space to the "skeleton" generated by keypoint pairs; a decoder that leverages the grid heatmap and the information in each layer of the encoder to reconstruct point clouds.

### 3.1 Encoder: Keypoint Detection

In Key-Grid, each keypoint is regarded as the weighted sum of all the points in the point cloud. Given an input point cloud $\mathbf{X} \in \mathbb{R}^{N \times 3}$, the goal of the encoder is to produce a weight matrix $\mathbf{W} \in \mathbb{R}^{K \times N}$, such that the matrix multiplication of $\mathbf{W}$ and $\mathbf{X}$ directly gives the predicted $K$ keypoints $\mathbf{K} \in \mathbb{R}^{K \times 3}$:

$$\mathbf{K} = \mathbf{W} \cdot \mathbf{X} \tag{1}$$

To be more specific, the encoder consists of $L$ PointNet++ [22] layers. The $i$-th PointNet++ layer in the encoder generates a hierarchically down-sampled point cloud $\mathbf{X}_{\text{enc}}^{(i)} \in \mathbb{R}^{N \times 3}$ and its corresponding feature $\mathbf{F}_{\text{enc}}^{(i)} \in \mathbb{R}^{N \times F}$, where $i \in \{1, 2, ..., L\}$. The last-layer feature passing through a Softmax activation function gives the weight matrix:

$$\mathbf{W} = \text{Softmax}(\mathbf{F}_{\text{enc}}^{(L)}) \tag{2}$$

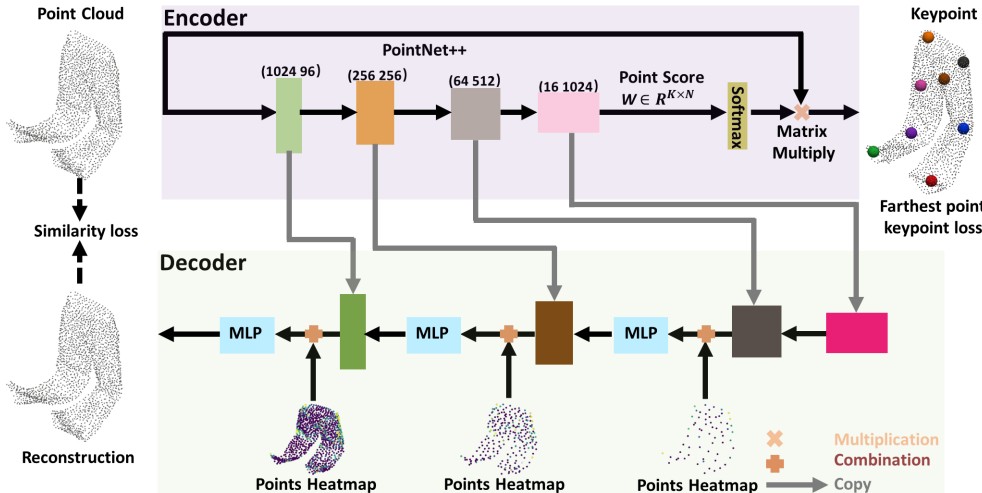

Figure 2: **Pipeline of Key-Grid.** In the encoder section, given a point cloud, we detect the keypoints by utilizing the PointNet++. Then, we utilize the detected keypoints to form the grid heatmap. In the decoder section, we use each layer of the PointNet++ and the grid heatmap to reconstruct the input point cloud. "MLP" stands for multi-layer perceptron, which contains Batch-norm and ReLU.

Additionally, following the practice in Skeleton Merger [26], the encoder outputs an additional head to predict the weights of $C_2^K = K(K-1)/2$ skeleton segments, i.e., the edges between each pair of keypoints. Formally, $(\mathbf{k}_i, \mathbf{k}_j)$ denotes the skeleton segment connecting the keypoints $\mathbf{k}_i \in \mathbb{R}^3$ and $\mathbf{k}_j \in \mathbb{R}^3$, and $s_{ij}$ denotes the weight of the skeleton segment $(\mathbf{k}_i, \mathbf{k}_j)$.

### 3.2 Grid Heatmap: Dense 3D Feature Map

The proposed grid heatmap is a dense 3D feature map designed to densely represent the 3D shape only through the information from the predicted keypoints. Ideally, for describing the object in an implicit manner similar to the Occupancy Networks [17], the feature on each grid is desired to reflect the distance from the grid point to the 3D object shape. Since the ground-truth object shape is the reconstruction target in our problem setting, we need to find ways to approximate the object shape using the predicted keypoints. In Key-Grid, we adopt the skeleton [26] approximation, where the object shape is represented by the weighted connected lines of all the keypoint pairs. For each individual grid point, we calculate the distances from this point to all the skeleton segments, and take the maximum of these distances as the feature of this grid. Intuitively, this gives a dense feature field whose value is the smallest at the geometric center of the object, and gradually increases along with the grid point coordinate moving outside the outline of the object shape. In the following, we illustrate the detailed procedures for establishing such a grid heatmap.

To begin with, we uniformly sample a 3D array of grid points $\mathbf{P} \in \mathbb{R}^{M \times M \times M \times 3}$ in the normalized cubic 3D space, where $M$ denotes the number of points we sample on each side of the cube. In Key-Grid, we set $M = 16$, giving 4096 gird points. Next, we calculate the distance between the grid points and all the line segments in the skeleton. When the projection of grid point onto the skeleton line falls in the range of the skeleton segment, the distance is defined as the distance between the grid point and the projection point. Otherwise, the distance is directly defined as the distance between the grid point and the nearest endpoint of the skeleton segment. Formally, the distance $d_{ij}(\mathbf{p})$ between a grid point $\mathbf{p} \in \mathbb{R}^3$ and a skeleton segment $(\mathbf{k}_i, \mathbf{k}_j)$ connecting the keypoints $\mathbf{k}_i$ and $\mathbf{k}_j$ is defined as:

$$d_{ij}(\mathbf{p}) = \begin{cases} \|\mathbf{p} - \mathbf{k}_i\|_2 & \text{if } t \leq 0 \\ \|\mathbf{p} - ((1-t)\mathbf{k}_i + t\mathbf{k}_j)\|_2 & \text{if } 0 < t < 1 \\ \|\mathbf{p} - \mathbf{k}_j\|_2 & \text{if } t \geq 1 \end{cases} \tag{3}$$

where

$$t = \frac{(\mathbf{p} - \mathbf{k}_i) \cdot (\mathbf{k}_j - \mathbf{k}_i)}{\|\mathbf{k}_i - \mathbf{k}_j\|_2^2} \in \mathbb{R} \tag{4}$$

Then, the feature of each grid point $D(\mathbf{p})$ is the maximum of the weighted distances from this point to all the skeleton segments:

$$D(\mathbf{p}) = \max_{ij} \left\{ s_{ij} \exp\left(d_{ij}^2(\mathbf{p})/\sigma^2\right) \right\} \tag{5}$$

where $s_{ij}$ refers to the learnable weight of the skeleton segment $(\mathbf{k}_i, \mathbf{k}_j)$ produced by the encoder, and $\sigma$ is a fixed hyperparameter. And the grid heatmap $\mathbf{H}$ is the 3D array consisting of the features of all the grid points:

$$\mathbf{H} = (D(\mathbf{P}_{xyz}))_{x,y,z=1,2,...,M} \in \mathbb{R}^{M \times M \times M \times 1}, \tag{6}$$

where $\mathbf{P}_{xyz} \in \mathbb{R}^3$ denotes the extracted grid point coordinate from $\mathbf{P}$ indexed by $(x, y, z)$. Conceptually, the grid heatmap $\mathbf{H}$ contains the complete geometric information of the keypoints $\mathbf{K}$. As a dense and continuous feature field rather than a set of sparse points, the grid heatmap is expected to force the encoder to predict precise keypoints and bring benefits to the reconstruction process.

In the grid heatmap, we measure the value of $D(\mathbf{p})$ using the maximum point-to-skeletons distance instead of minimum distance because the maximum distance can better distinguish the spatial locations, especially what is inside or outside an object. For instance, Figure 3 shows that if we choose the minimum distance between the sampled grid points and the keypoints, the grid point $p_1$ inside the pants will have the same value $d_{min}$ as the grid point $p_0$ outside the pants. However, if we take the maximum value $d_{max}$, the grid point $p_1$ inside the pants will have a smaller value than the grid point $p_0$ outside the pants.

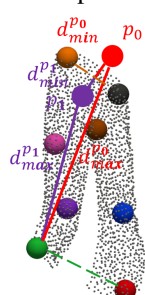

Figure 3: Example of distance definition on the grid heatmap.

### 3.3 Decoder: Point Cloud Reconstruction

As an inverse process of the encoder, the decoder tries to reconstruct the entire point cloud by gradually augmenting finer geometric details in a hierarchical manner. Unlike previous methods [26; 9] where the keypoint-related 3D structures are used only once as the input of the encoder, we propose to integrate increasingly finer grid heatmap features with the layers of the decoder. In addition, the features in the encoder are directly copied and fused into the corresponding layer in the decoder in a U-Net [23] fashion to further assist the reconstruction process.

Formally, the feature of the $(L - i + 1)$-th layer in the decoder is composed by the following three components:

$$\mathbf{F}_{dec}^{(L-i+1)} = \mathbf{H}(\mathbf{X}_{enc}^{(i)}) \oplus \mathbf{F}_{enc}^{(i)} \oplus \text{Proj}(\mathbf{F}_{dec}^{(L-i)}, \mathbf{X}_{enc}^{(i-1)}, \mathbf{X}_{enc}^{(i)}), \tag{7}$$

where $\mathbf{H}(\mathbf{X}_{enc}^{(i)}) \in \mathbb{R}^{N \times 1}$ denotes the extracted grid features indexed by $\mathbf{X}_{enc}^{(i)} \in \mathbb{R}^{N \times 3}$, $\mathbf{F}_{enc}^{(i)} \in \mathbb{R}^{N \times F}$ denotes the corresponding feature of the same number of points copied from the encoder, $\text{Proj}(\mathbf{F}_{dec}^{(L-i)}, \mathbf{X}_{enc}^{(i-1)}, \mathbf{X}_{enc}^{(i)})$ denotes the feature projected from the former layer of the decoder, and $\oplus$ denotes element-wise concatenation. Note that $\mathbf{H}(\mathbf{X}_{enc}^{(i)})$ and $\mathbf{F}_{enc}^{(i)}$ are already aligned on the first dimension and thus ready for concatenation, the remaining challenge is how to design a feature projection mechanism $\mathbf{F}_{targ} = \text{Proj}(\mathbf{F}_{ori}, \mathbf{X}_{ori}, \mathbf{X}_{targ})$ so that the former-layer features can be aligned with the current-layer features.

To solve this problem, we propose to represent the target feature $\mathbf{F}_{targ}$ as the weighted sum of the spatially neighboring features in the original feature $\mathbf{F}_{ori}$. More specifically, for every point coordinate $\mathbf{x} \in \mathbb{R}^3$ in the target point cloud $\mathbf{X}_{targ}$, we find its $N_{neig}$ neighbors in the original point cloud $\mathbf{X}_{ori}$:

$$\mathbf{X}_{neig}(\mathbf{x}|\mathbf{X}_{ori}) = \left\{ \mathbf{X}_{ori}^{(1)}, \mathbf{X}_{ori}^{(2)}, ..., \mathbf{X}_{ori}^{(N)} \right\}. \tag{8}$$

The target feature of the coordinate $\mathbf{x}$ is defined as:

$$\mathbf{F}_{targ}(\mathbf{x}) = \frac{\sum_{\acute{\mathbf{x}} \in \mathbf{X}_{neig}(\mathbf{x}|\mathbf{X}_{ori})} \omega(\mathbf{x}, \acute{\mathbf{x}}) \mathbf{F}_{ori}(\acute{\mathbf{x}})}{\sum_{\acute{\mathbf{x}} \in \mathbf{X}_{neig}(\mathbf{x}|\mathbf{X}_{ori})} \omega(\mathbf{x}, \acute{\mathbf{x}})}, \tag{9}$$

where $\omega(\mathbf{x}, \acute{\mathbf{x}})$ denotes the inverse of the squared distance between $\mathbf{x}$ and $\acute{\mathbf{x}}$:

$$\omega(\mathbf{x}, \acute{\mathbf{x}}) = \frac{1}{\|\mathbf{x} - \acute{\mathbf{x}}\|_2^2}. \tag{10}$$

### 3.4 Training Objectives

Our approach adopts an end-to-end method called Key-Grid to identify the keypoints, by leveraging the grid heatmap to reduce the discrepancy between the reconstructed point cloud and the input point cloud. In this section, we will introduce two types of loss function to optimize Key-Grid.

**Similarity loss.**   The common way to evaluate the similarity between a reconstructed point cloud $\mathbf{X}_r$ and a target point cloud $\mathbf{X}$ is by employing the Chamfer distance metric [9; 48]. By minimizing the Chamfer distance between them, we can optimize the reconstruction process to closely match the target point cloud, thereby achieving a higher level of similarity between the two point clouds. Thus, we utilize the Chamfer distance between the reconstructed point cloud $\mathbf{X}_r$ and the target point cloud $\mathbf{X}$ to approximate this similarity loss $\mathcal{L}_{sim}$.

**Farthest point keypoint loss.**   To ensure that the aligned keypoints distribute well on the surface of objects and represent the geometric structure of objects, we use the farthest point keypoint loss [9] $\mathcal{L}_{far}$ to allocate the keypoints. Firstly, we randomly select an initial point from the point cloud $\mathbf{X}$. Then, at each iteration, it selects the point that is farthest from all the previously selected points. This process continues until $J$ points have been selected. Finally, we can obtain different sets of sampled points $\mathbf{Q} = \{\mathbf{q}_1, \mathbf{q}_2, \cdots, \mathbf{q}_J\} \in R^{J \times 3}$. These sampled farthest points $\mathbf{Q}$ are regarded as a prior estimation of the distribution of keypoints. We define this loss as minimizing the Chamfer Distance between the predicted keypoints $\mathbf{K}$ and the sampled points $\mathbf{Q}$.Our overall loss function is

$$\mathcal{L}_{over} = \alpha_{sim}\mathcal{L}_{sim} + \alpha_{far}\mathcal{L}_{far} \tag{11}$$

where $\alpha_{far}$ and $\alpha_{sim}$ are scalar loss coefficients.

## 4   Experiments

In this section, we compare the performance of Key-Grid over the existing SOTA approaches on both rigid-body and deformable object datasets. Meanwhile, we conduct the robustness analysis, ablation studies, and show Key-Grid can be easily extended to an SE(3)-equivariant version.

### 4.1   Setup

**Datasets.**   We use the **ShapeNetCoreV2** and the **ClothesNet** datasets [2; 49] to evaluate the performance of Key-Grid. For the **ShapeNetCoreV2** dataset [2], it contains 51,300 rigid-body objects of 55 different categories. In this paper, we only use categories that are manually annotated with semantic correspondence labels in the KeypointNet dataset [42]. For the **ClothesNet** dataset [49], we take three types of deformations on different type of clothing objects: dropping, dragging, and folding. For each deformation of different garment categories, there are 128 samples during the deformation process. We test the performance of Key-Grid on real-world 3D scans of clothing from the **Deep Fashion3D V2** dataset [6] and conduct the additional experiments on **SUN3D** dataset [38] to further illustrate Key-Grid capability in coping with real and large-scale dataset.

**Baselines.**   We compare Key-Grid with the current SOTA approaches: **KeypointDeformer (KD)** [9], **Skeleton Merger (SM)** [26] and **SC3K** [51] that all detect the 3D keypoints in an unsupervised way. Compared to SM [26], SC3K [51], and Key-Grid, KD [9] not only requires point clouds as input but relies on object meshes to perform shape transformations from the source shape to the target shape. Therefore, regarding KD [9] as a baseline is actually unfair to our method. However, since it is the current SOTA method for keypoint detection on deformable objects, we choose it as a baseline to compare with our method.

**Evaluation metrics.**   For the ShapeNetCoreV2 dataset [2], we use the **Dual Alignment Score (DAS)** [26] to assess the degree of keypoints semantic consistency for each category. Meanwhile, to verify the accuracy of detected keypoint locations, we compute the **mean Intersection over Union (mIoU)** metric [33] with a threshold of 0.1 to evaluate the difference between the detected keypoints and the ground truth provided by the KeypointNet dataset [42]. For the ClothesNet dataset [49], due to the absence of manually annotated keypoints, we only use DAS to evaluate the semantic consistency of keypoint detection on objects with three type of deformations. Higher values for both metrics correspond to better model performance.

### 4.2   Results

For the ShapeNetCoreV2 dataset [2], we adopt Key-Grid and other baselines to detect ten keypoints and use the DAS and mIoU to evaluate their performances on the 13 categories of rigid-body objects in the Table 1. For objects with straight-line geometrical structures such as "Airplane", "Vessel", "Knife", and "Guitar", skeletons formed by connecting keypoints easily represent their

| Category | DAS↑ | | | | mIoU↑ | | | |
|---|---|---|---|---|---|---|---|---|
| | **KD** | **SM** | **SC3K** | **Key-Grid (Ours)** | **KD** | **SM** | **SC3K** | **Key-Grid (Ours)** |
| Airplane | 73.4 | 77.8 | 82.9 | **84.7** | 53.3 | 73.5 | **74.9** | 74.4 |
| Bed | 69.5 | 61.3 | 64.9 | **70.6** | 40.7 | **59.9** | 19.9 | 51.2 |
| Bottle | 60.3 | 59.9 | 62.7 | **66.2** | 34.0 | 36.5 | 38.0 | **38.7** |
| Cap | 56.7 | 53.1 | **59.7** | 58.5 | 9.3 | 15.5 | 11.3 | **17.6** |
| Car | 80.2 | 79.5 | 75.2 | **84.5** | 53.8 | 53.3 | 33.2 | **56.1** |
| Chair | 84.3 | 76.8 | 87.0 | **93.4** | 62.0 | 60.0 | 39.8 | **65.4** |
| Guitar | 62.9 | 63.2 | 65.7 | **69.1** | 52.6 | 49.7 | **68.0** | 56.2 |
| Helmet | 55.3 | 57.0 | 58.6 | **62.5** | 13.5 | 16.7 | 17.6 | **39.2** |
| Knife | 59.8 | 60.4 | **63.0** | 60.9 | 38.1 | 45.0 | **47.9** | 46.8 |
| Motorbike | 61.5 | 57.8 | 59.4 | **63.7** | 45.7 | 39.7 | 39.7 | **48.2** |
| Mug | 69.8 | 67.2 | 75.3 | **79.4** | 12.6 | 23.6 | 18.4 | **24.3** |
| Table | 72.6 | 70.0 | 76.0 | **78.6** | 62.0 | 60.9 | 46.4 | **66.5** |
| Vessel | 73.9 | 72.4 | **76.0** | 74.6 | 51.2 | **58.9** | 47.5 | 52.4 |
| Average | 67.7 | 65.8 | 69.7 | **72.8** | 40.7 | 45.6 | 38.7 | **49.0** |

Table 1: **Comparative DAS and mIoU score: Key-Grid vs. State-of-the-Art Approaches on the ShapeNetCoreV2 dataset.** ↑ means better performance. The results are calculated for **10** keypoints and the DAS value of SM and SC3K is reported in [51] and [26]. The mIoU values of each method are the results we reproduced based on their official code. **Colorbox** and underlined respectively represent the first and second best performance in all tables of this paper.

| Category | Drop Clothes | | | | Drag Clothes | | | |
|---|---|---|---|---|---|---|---|---|
| | **KD** | **SM** | **SC3K** | **Key-Grid (Ours)** | **KD** | **SM** | **SC3K** | **Key-Grid (Ours)** |
| Hat | 96.7 | 77.7 | 55.7 | **100.0** | 50.0 | 42.7 | 42.7 | **51.5** |
| Long Pants | 57.5 | 46.4 | 77.7 | **83.7** | 87.5 | 90.6 | 81.3 | **99.0** |
| Jacket | 77.5 | 44.6 | 78.6 | **90.2** | 39.6 | 66.7 | 45.9 | **67.7** |
| Long Dress | 70.8 | 73.2 | 69.6 | **75.9** | 61.5 | **69.8** | 50.0 | 61.5 |
| Short Dress | 90.2 | 72.6 | 57.1 | **91.1** | 70.8 | 66.7 | 57.3 | **76.0** |
| Mask | 94.3 | 95.5 | 91.7 | **100.0** | 29.2 | 43.8 | 26.0 | **47.9** |
| Polo | 65.8 | 48.2 | 64.3 | **73.2** | 55.2 | 44.8 | 38.5 | **62.8** |
| Scarf | 87.5 | 75.0 | 54.5 | **91.1** | 51.1 | **65.6** | 58.3 | 59.3 |
| Shirt | 74.1 | 55.4 | 84.8 | **87.5** | 59.4 | 45.8 | 45.8 | **60.2** |
| Short Pants | 84.2 | 86.6 | 28.6 | **100.0** | 72.9 | 65.6 | 67.8 | **81.3** |
| Skirt | 89.4 | 37.5 | 92.9 | **95.5** | 52.1 | 53.1 | 21.9 | **59.3** |
| Tie | 79.2 | **87.5** | 37.5 | 79.5 | 55.2 | 56.3 | 33.3 | **64.6** |
| Vest | 96.7 | 76.8 | 96.4 | **100.0** | 67.7 | 50.0 | 62.5 | **89.6** |
| Average | 81.8 | 67.5 | 68.4 | **89.8** | 57.9 | 58.6 | 48.6 | **67.7** |

Table 2: **Comparative DAS score: Key-Grid vs. State-of-the-Art Approaches on the ClothesNet dataset.** We demonstrate the DAS values for **8** keypoints recognized by different methods on 13 types of clothing under the dropping and dragging deformations.

geometrical structures, which is why SM outperforms Key-Grid for these categories on the mIoU metric. However, Key-Grid's performance on these categories is comparably optimal, achieving higher average scores for both DAS and mIoU compared to other baselines. Additionally, we present various supervised networks and self-supervised methods to compare Key-Grid on the KeypointNet dataset [42] based on mIoU metric in Table 3. Several standard networks, PointNet [21], SpiderCNN [39], and PointConv [36], are trained to predict keypoint probabilities in a supervised manner. We can observe that Key-Grid demonstrates superior accuracy in keypoint localization compared to other self-supervised methods and outperforms some supervised approaches that utilize PointNet and SpiderCNN as backbones.

| | Airplane | Chair | Car | **Average** |
|---|---|---|---|---|
| PointNet [21] | 45.4 | 23.8 | 15.3 | 28.2 |
| SpiderCNN [39] | 55.0 | 49.0 | 38.7 | 47.6 |
| PointConv [36] | **93.5** | **86.0** | **83.6** | **87.0** |
| SM [26] | 79.4 | 68.4 | 63.2 | 70.3 |
| SC3K [51] | 82.7 | 38.5 | 34.9 | 52.0 |
| **Key-Grid** | 80.9 | 75.2 | 69.3 | 75.1 |

Table 3: **Comparative mIoU score: Key-Grid vs. Supervised Keypoint Approaches on the KeypointNet dataset.** The results of the mIoU score are calculated for **10** keypoints on the KeypointNet dataset. The results of supervised keypoint detection method are reported in [26].

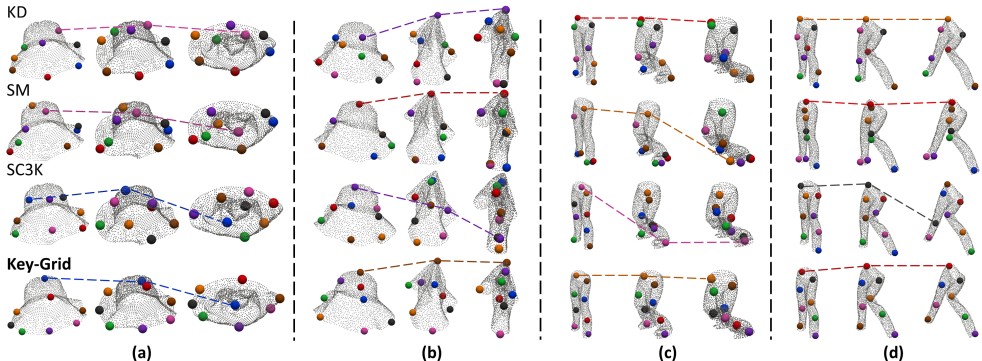

Figure 4: **Different methods on the Hat and Long Pant categories during the dropping and dragging processes.** **(a)** and **(b)**: Keypoint detection of Hat under the dropping and dragging deformation. **(c)** and **(d)**: Keypoint detection of Long Pant under the dropping and dragging deformation. We use lines to connect keypoints of the same color, representing the positional changes of the same keypoints in the deformation process of the objects.

| Category | Fold Clothes | | | | | | | |
| | Normal Placement | | | | SE(3) Transformation | | | |
| | KD | SM | SC3K | Key-Grid (Ours) | KD | SM | SC3K | Key-Grid (Ours) |
|---|---|---|---|---|---|---|---|---|
| Shirt | 81.6 | 79.5 | 53.6 | **92.0** | 80.4 | 75.9 | 51.9 | **90.2** |
| Pant | 72.4 | 71.4 | 32.1 | **100.0** | 70.5 | 69.6 | 28.6 | **98.2** |

Table 4: **Comparison of DAS values for Folded Clothes under Normal Placement and SE(3) Transformation.** For deformations with large changes, such as folding, Key-Grid has a more noticeable advantage whether the clothes are placed normally or undergo SE(3) transformation.

For the ClothesNet dataset [49], we aim to make the eight detected keypoints maintaining semantic consistency in objects under the deformation process of the same category. Table 2 shows Key-Grid performs better on the deformations of dropping and dragging than other methods. Key-Grid outperforms other baselines in the drop deformation for all categories, except for "Tie", while also demonstrating superior performance in the drag deformation for all categories, except for "Long Dress" and "Scarf". Additionally, we also show the semantic consistency of keypoints identified by different methods during the folding process of shirts and pants in Table 4. Table 4 illustrates that for objects with significant deformations, such as folding, Key-Grid significantly outperforms other methods in terms of keypoint semantic consistency. Figure 4 and Figure 5 show the visualization results of keypoint detection under three deformations of clothing using different methods. In Figure 4, for the dropping and dragging deformation process on the "Hat" and "Long Pant" categories, compared to other methods, keypoints identified by Key-Grid distribute evenly on objects and keep great semantic consistency. The keypoint positions do not change with the deformation of the objects. In Figure 5(a) and (b), we can observe that when facing folding deformation, keypoints detected by KD and SM have redundancy phenomenon, which means multiple keypoints are identified at the same location. Compared to the SC3K method, keypoints identified by Key-Grid not only capture essential geometric details of deformable objects but also ensure semantic coherence throughout the folding process. Figure 5(c) shows the grid heatmaps are evidently better at accurately reflecting the geometric structure of the folded clothes than the skeleton structures proposed in SM. Thus, we think that precise representation of the object structure using keypoints forces the network to generate precise keypoints, which makes Key-Grid perform better than previous methods.

For the Deep Fashion3D V2 dataset [6], we select three shirts with different deformations for keypoint recognition. Key-Grid successfully learns eight semantically consistent 3D keypoints for these objects, as shown in Figure 5(d), even with obvious deformations in the sleeves of the clothes. The hyperparameters used in this experiment are consistent with those employed on the ClothesNet dataset.

For the SUN3D dataset [38], Figure 5(d) presents the visualization results of 20 keypoints on the different local geometric scenes identified by Key-Grid. Despite the influence of real-distributed noise and occlusion on keypoint detection results, Key-Grid can still recognize keypoints with important geometric information in real-level data, such as corners of buildings and center positions of scenes.

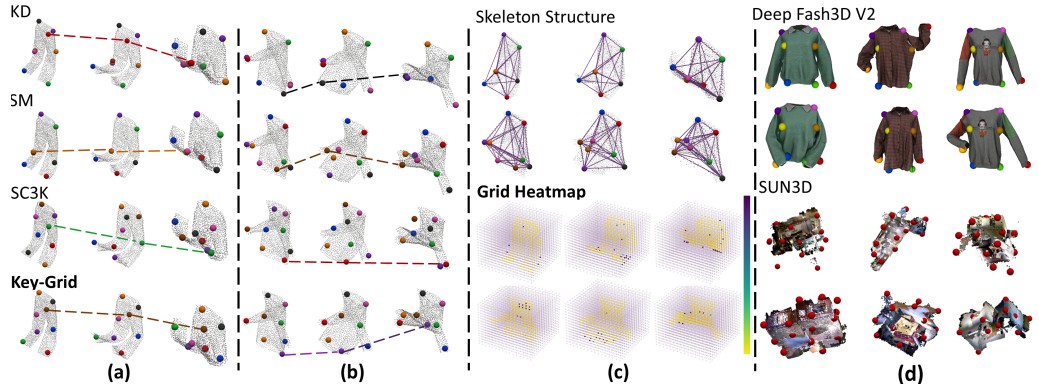

Figure 5: **Keypoints detected on the Fold Clothes, the Deep Fash3D V2 dataset and the SUN3D dataset.** (a) and (b): Eight keypoints identified by different methods during the folding process of clothes. The lines connect the keypoints with the same color, which means the positions of these keypoints change in the deformation process. (c): Grid Heatmaps and Skeleton Structures on the fold clothes. In the skeleton structures, we use purple dots to connect the keypoints identified by SM to construct the skeleton. In the grid heatmap, we use colors to represent the values of $D(\mathbf{p})$, with yellow indicating smaller values. The yellow dots capture the geometric structure of the folded clothes. (d): Keypoints detected by Key-Grid on the Deep Fash3D V2 and the SUN3D dataset.

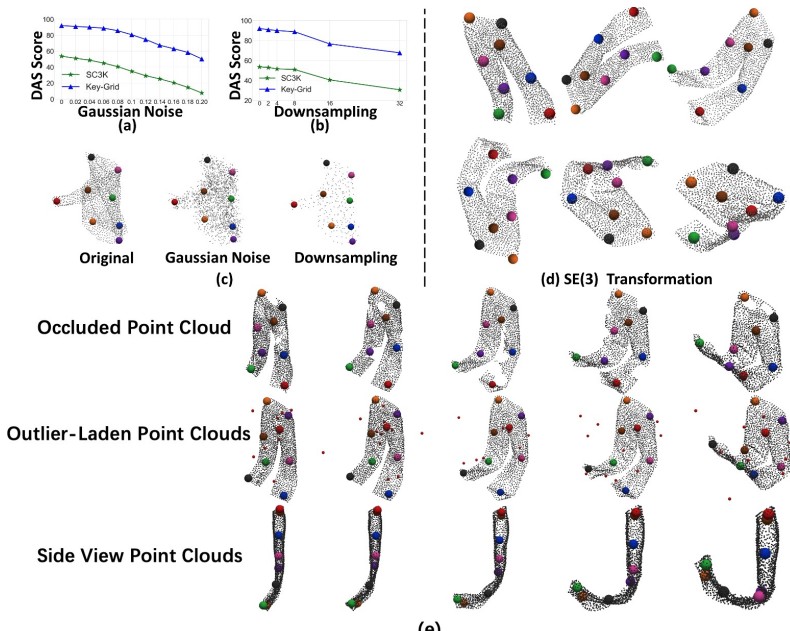

Figure 6: **Robustness Analysis of Key-Grid and Visualization Results of the SE(3)-Equivariant Keypoints.** (a) and (b): DAS value under Gaussian noise and downsampling. (c): Visualization results of Key-Grid under these situations. The Gaussian noise scale is 0.06 and the downsample rate is 8x, respectively. (d): Keypoints on the folded pants undergoing SE(3) transformations. (e):Keypoint detection on the on occluded, side view, and outlier-laden point clouds.

## 4.3 Robustness Analysis

In this section, we focus on evaluating the robustness of Key-Grid on the noisy and down-sampled point clouds of folded shirts. To obtain the noisy point clouds, we add Gaussian noise with varying variances to the point clouds. And we utilize the Farthest Point Sampling method to downsample the original point clouds, which has also been used in previous works [19; 47].

Figure 6(a), (b) and (c) show the DAS and visualization results of detected keypoints for the noisy and down-sampled point clouds. The results indicate that both SC3K and Key-Grid experience a decrease in DAS as the noise level increases. Meanwhile, when downsampling the original point cloud by a

factor of 16, the DAS of Key-Grid and SC3K will significantly decrease. However, when Key-Grid is subjected to significant noise interference and high-level downsampling, the DAS of Key-Grid is still better than SC3K, which has not been subjected to any interference. The visualization results also show that keypoints detected by Key-Grid exhibit strong robustness even when subjected to Gaussian noise or downsampling.

Additionally, we also present the visualization of keypoints identified from occluded objects, outlier-laden objects, and side views of objects under the folding deformation in Figure 6(e). Key-Grid shows robust performance on both occluded point clouds and those captured from side views. Meanwhile, even when point clouds contain outliers, keypoints detected by Key-Grid maintain strong semantic consistency. Thus, we can conclude that Key-Grid effectively identifies keypoints in comparatively low-quality or partial point clouds, similar to its performance with complete point clouds.

### 4.4 SE(3)-Equivariant Keypoints

It is significant that Key-Grid is capable of identifying keypoints of objects undergoing SE(3) transformations. We analyze the capability of Key-Grid to identify keypoints of objects undergoing SE(3) transformations. However, if using a SPRIN module [43] which is a SOTA SE(3)-invariant backbone to replace PointNet++ [22] directly, the training process of Key-Grid does not converge. Thus, to achieve Key-Grid to accurately identify the keypoints of objects undergoing SE(3) transformations, we regard Key-Grid as the teacher network of USEEK [40]. The student network of USEEK is the SPRIN module. Figure 6(d) shows that the keypoints of folded pants recognized by USEEK are invariant under an assortment of random SE(3) transformations. We also consider other baselines as the teacher model of USEEK. Table 4 demonstrates that Key-Grid outperforms other baselines in terms of identifying the SE(3)-Equivariant keypoints with semantic consistency.

### 4.5 Ablation Studies

**Strategy of decoder.**   In the decoder section, Key-Grid combines the information from each layer of the encoder with the grid heatmap to reconstruct the original point cloud. In order to illustrate the importance of encoder information and grid heatmap for the reconstruction process, Key-Grid respectively uses one of these two strategies in the decoder section. Table 5 shows that when using both input streams to reconstruct the point cloud, the keypoints detected by Key-Grid exhibit better semantic consistency.

**Loss ablation.**   To emphasize the importance of each loss, we conduct the evaluation of the proposed approach by systematically excluding each loss individually. The results are illustrated in Table 5. We can observe that the DAS of Key-Grid decreases when any of the loss functions is excluded from the training process. The similarity loss contributes comparatively low, but regardless of whether Key-Grid is applied on the ClothesNet dataset or the ShapeNetCoreV2 dataset, it still can improve the semantic consistency of the keypoints identified by Key-Grid. The contribution of the farthest point keypoint loss is comparatively higher than the similarity loss. Without the farthest point keypoint loss, the DAS of Key-Grid will decrease significantly.

| | ShapeNetCoreV2 | | ClothesNet | |
|---|---|---|---|---|
| | Airplane | Chair | Folded Shirt | Folded Pant |
| **Key-Grid** | **84.7** | **93.4** | **92.0** | **100.0** |
| No Encoder Information | 73.4 | 82.6 | 81.7 | 90.8 |
| No Grid Heatmap | 73.2 | 81.7 | 79.5 | 87.5 |
| No Similarity | 71.5 | 78.6 | 77.7 | 84.5 |
| No Farthest Point | 15.6 | 13.7 | 17.9 | 17.0 |

Table 5: DAS of ablation study on the ShapeNetCoreV2 and ClothesNet dataset.

## 5   Conclusion

In this paper, we propose Key-Grid that detects keypoints for both rigid-body and deformable objects. It uses the detected keypoints to build grid heatmap and incorporate it into the reconstruction process of point clouds. We evaluate the quality of keypoints on multiple datasets and analyze the robustness of Key-Grid. Meanwhile, we embed the Key-Grid into the USEEK framework to produce SE(3)-equivariant keypoints. Extensive experiments shows that Key-Grid can detect the keypoints with great semantic consistency and precise location.

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

# Appendix Materials

## A    Implementation Details

We use the PyTorch framework [7] to train our method. The entire network is optimized on the Adam optimizer [10] with a learning rate of 0.1. For all the experiments, the batch size is set to 8. Our method is trained for 100 epochs, and for the first 20 epochs of training, we do not use a similarity loss function which measures the similarity between the reconstructed point clouds and the original point clouds.

For all the experiments, the input point cloud contains $N = 2048$ points. The number of encoder layers $L$ in the PointNet++ network [22] is 4. For the ShapeNetCoreV2 dataset [2], we set the number of keypoints $K$ to be 10. Thus, when calculating the farthest point keypoint loss function, we choose the number of the farthest points $J$ as 14. For the ClothesNet [49] and Deep Fash 3D dataset [6], we set $K$ and $J$ as 8 and 12, respectively. The hyperparameter $K$ and $J$ are 20 and 24 in the SUN3D dataset [38]. In the process of building the grid heatmap, the number of points $M$ uniformly selected on each edge of the cubic space is 16. For the decoder, the hyperparameter $N_{\text{neig}}$ for the number of neighboring features applied in aligning the feature from the former layer to the current layer is set to 3. In the loss function, $\alpha_{far}$ and $\alpha_{sim}$ are both set to 1.

## B    Efficiency of Key-Grid

Table 6 presents the time and memory consumption of Key-Grid and SM when inferring a batch comprising 32 samples on a single 1080 Ti GPU. According to Table 6, Key-Grid achieves equal efficiency to the baseline method (SM) in terms of inference speed but consumes an acceptably larger memory size.

| Method | Memory (MiB) | Time (s) |
|--------|--------------|----------|
| SM     | **4531**     | 0.6      |
| Ours   | 6081         | **0.6**  |

Table 6: Analysis of Key-Grid efficiency.

## C    Visualization Results

In this section, we primarily present additional visualization results about the experimental part to illustrate the effectiveness of Key-Grid.

**Number of keypoints.**    We vary the number of unsupervised keypoints discovered by Key-Grid on the folded pants. Figure 7 shows that these different numbers of keypoints on the folded pants maintain good semantic consistency during the deformation process. And as the number of keypoints increases, we find that these keypoints do not overlap with each other, they still evenly distribute on the surface of the objects, and each keypoint represents the geometric information of the objects.

**Ablation study.**    We show the visualization results of the ablation study in Figure 8. From the Figure 8, we find that in the decoder section, if we only use the encoder information or the grid heatmap information to reconstruct the original point cloud, the positions of the keypoints on the folded pants are not stable. When we do not use the similarity loss function in the decoder section, the keypoints will be placed in the wrong prior positions and the network predicts the redundant keypoints. Moreover, when we do not apply the farthest point keypoint loss function to train the network, the performance of keypoint recognition will be poor, as the keypoints will cluster around the center of the object.

**Gaussian noise to point cloud.**    Figure 9 shows the visualization results of Key-Grid for different noisy point clouds, which add the Gaussian noise of different scales to the input point cloud. We can observe that even when Gaussian noise is added to the original point cloud with a magnitude of 0.08,

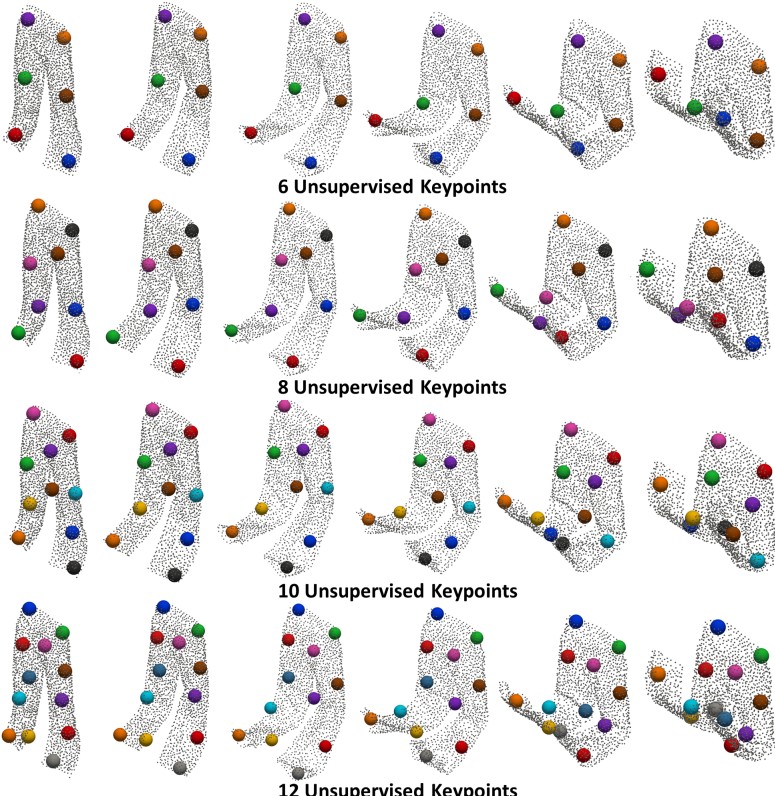

**6 Unsupervised Keypoints**

**8 Unsupervised Keypoints**

**10 Unsupervised Keypoints**

**12 Unsupervised Keypoints**

Figure 7: **Different number of detected keypoints on the folded pants.** Key-Grid identifies different numbers (6, 8, 10, 12) of keypoints on the folded pants, respectively.

the positions of the keypoints detected by Key-Grid in these noisy point clouds are still close to those found in the original point cloud. Thus, we can conclude that Key-Grid exhibits robustness when dealing with the noisy point clouds.

**Downsampling point cloud.** This paragraph presents the performance of Key-Grid on the downsampled point clouds shown in Figure 10. For decimating the original point cloud, we use the Farthest Point Sampling method for downsampling. We discover that Key-Grid successfully estimate the keypoints on the downsampled point clouds. When downsampling the original point cloud by a factor of 16, resulting in a point cloud with 128 sample points, except for the blue keypoints which will be shifted downwards, the positions of the other keypoints will remain close to their positions detected in the original point cloud.

**Deep Fash3D V2 dataset.** We show the additional visualization results of keypoints detected by Key-Grid on the Deep Fash3D V2 dataset [6]. Figure 13 illustrates that when faced with a wider variety of clothing cuff deformations, the keypoints detected by Key-Grid achieves great semantic consistency on the world scanned objects and represents the important information about these objects. Thus, we think that Key-Grid performs well in the world scanned objects.

**ShapeNetCoreV2 and ClothesNet dataset.** For ShapeNetCoreV2 [2] and ClothesNet [49] dataset, we present the visualization results of keypoint recognition in a wider range of object categories using SM and Key-Grid. In Figure 12, we separately present the visualization results of keypoint recognition on the ShapeNetCoreV2 dataset using SM and Key-Grid. Compared to the redundancy phenomenon that many keypoints locate in the same region generated by SM [26], the keypoints detected by Key-Grid distribute separately on the surface of the object, and these keypoints also can represent the important geometric information of the object. For the ClothesNet dataset, Figure 13 demonstrates the keypoint detection of SM [26] and Key-Grid in the deformation process of a wider range of object categories, including drop and drag deformations. In the deformation process, compared to SM [26],

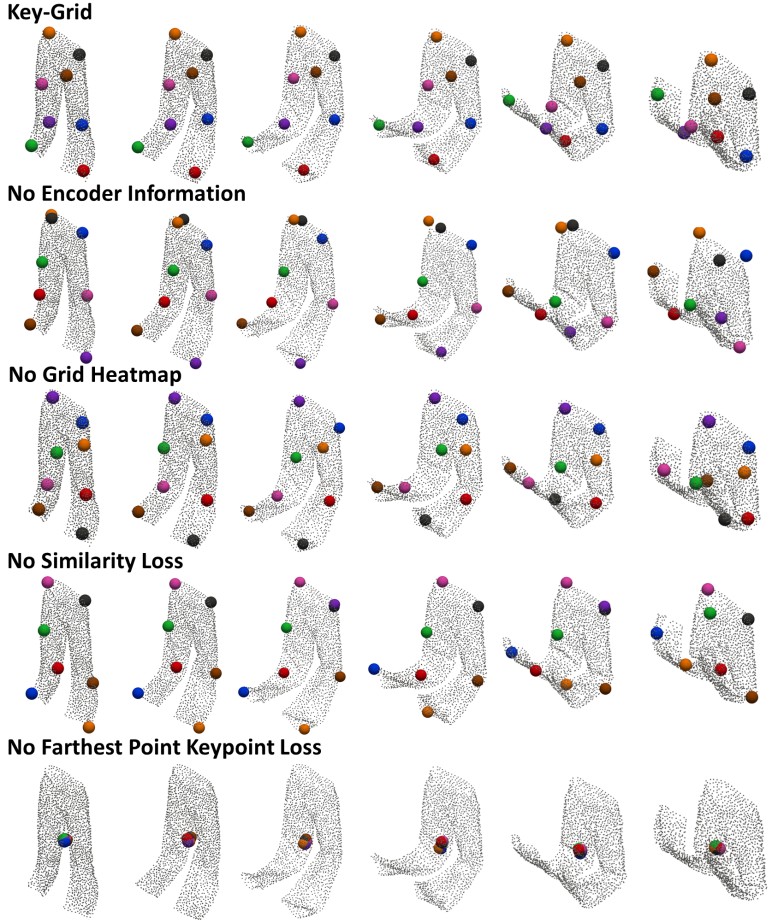

Figure 8: **Visualization results of the ablation study on the folded pants.**

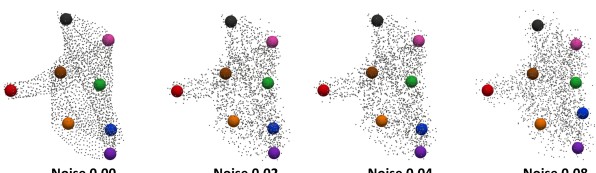

Figure 9: **Performance of Key-Grid on the noisy point clouds.** We indicate the level of Gaussian noise added underneath each noisy point cloud. "Noise 0.00" is the original point cloud.

the keypoints identified by our method keep good semantic consistency and distribute evenly on the surface of the object. Thus, for the rigid-body and deformable objects, Key-Grid exhibits great performance on keypoint detection.

# D   Video Results

In the supplementary materials, we also provide keypoint recognition results for the folding clothes in the ClothesNet dataset using different methods. The video we provide include the entire process of deformation for shirts and pants. From this video, we can conclude that Key-Grid identifies the keypoints that maintain good semantic consistency during the deformation process. These keypoints also carry the geometric feature information of the deformable objects.

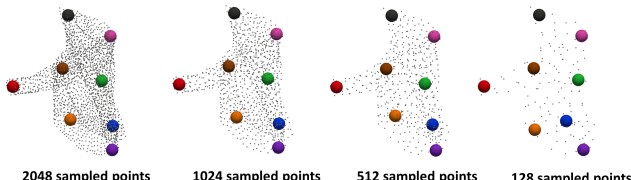

Figure 10: **Performance of Key-Grid on the downsampled point clouds.** The input point clouds are downsampled for different scales, as mentioned at the bottom of each point cloud. "2048 sampled points" is the input point cloud.

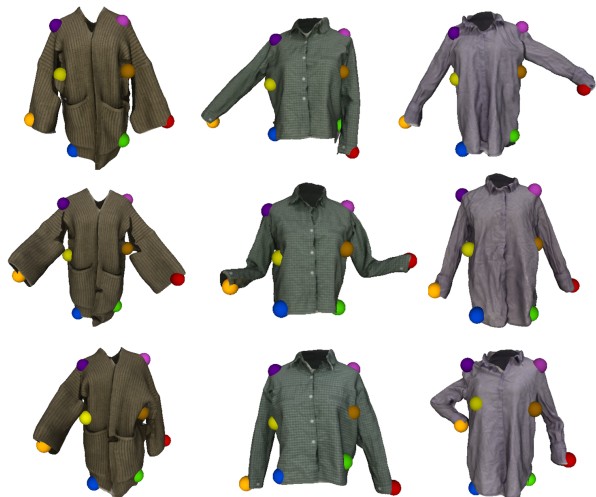

Figure 11: **Additional visualization results on the Deep Fash3D V2 dataset.** Key-Grid recognizes the keypoints in these three garments with different deformation.

## E    Limitation and Social Imapct

Following previous works, we need to manually set the total number of predicted keypoints before-hand. In future research, we will propose an adaptive keypoint detection method that can generate varying numbers of keypoints with accurate positions for different samples. Key-Grid predicts the keypoints on the publicly available datasets. Thus, we think Key-Grid has very limited potential negative societal impacts.

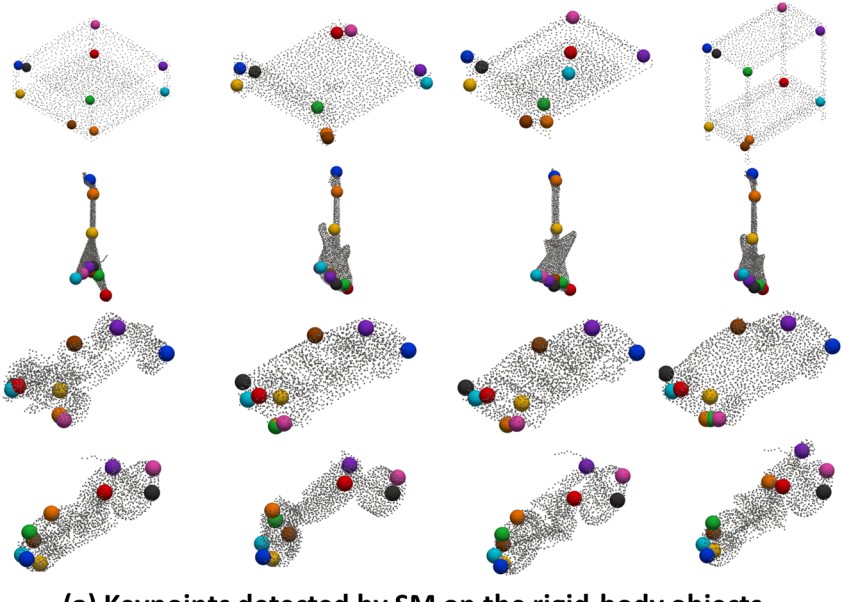

**(a) Keypoints detected by SM on the rigid-body objects**

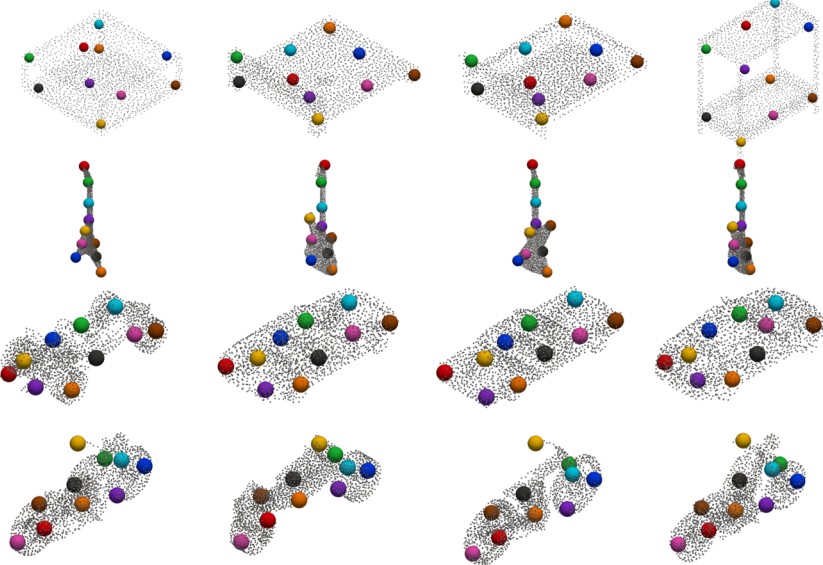

**(b) Keypoints detected by Key-Grid on the rigid-body objects**

Figure 12: **Keypoint detection on the ShapeNetCoreV2 dataset.** The keypoints are detected by SM and Key-Grid on the "Bed", "Guitar", "Car", and "Motorcycle" category of objects in the ShapeNetCoreV2 dataset. Each category shows four samples.

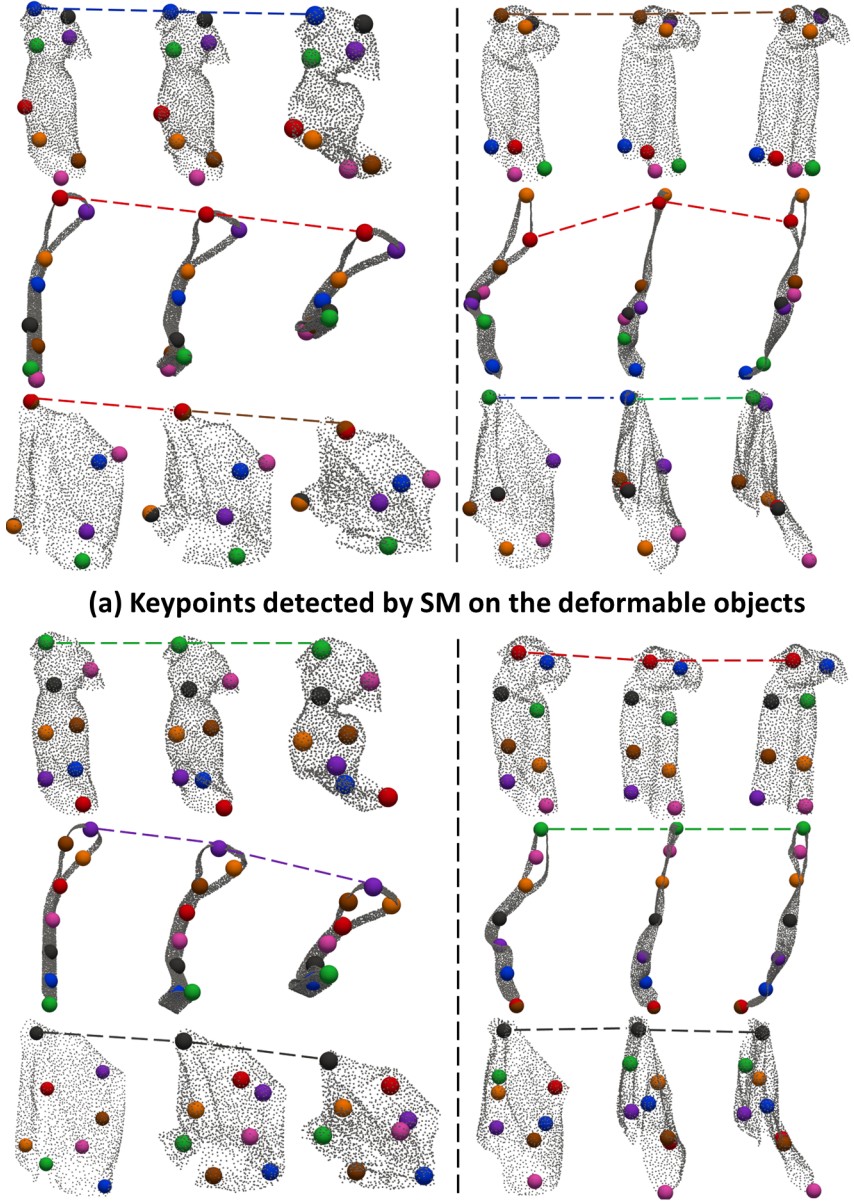

**(a) Keypoints detected by SM on the deformable objects**

**(b) Keypoints detected by Key-Grid on the deformable objects**

Figure 13: **Keypoint detection on the ClothesNet dataset.** For the drop and drag deformation processes, we select the "Long Dress", "Tie", and "Vest" categories of objects to visualize the keypoints identified by SM and Key-Grid. We use lines to connect the keypoints at the same position during the deformation process.

