# OpenReview forum: "Key-Grid: Unsupervised 3D Keypoints Detection using Grid Heatmap Features"
_NeurIPS.cc/2024/Conference — NeurIPS 2024 poster_

### Official Review · Reviewer_LZF7 · 2024-06-22

**Soundness:** 2
**Presentation:** 3
**Contribution:** 2
**Rating:** 4
**Confidence:** 2

**Summary:**

This paper presents a novel unsupervised network for keypoint detection. The method can be applied to handle both rigid and deformable objects. It follows an autoencoder framework where the encoder predicts keypoints and the decoder utilizes the generated keypoints to reconstruct the objects. The main contribution lies in the grid based representation which is suitable to capture the shifted geometric structure of the keypoints brought by deformations.

**Strengths:**

1 This paper presents an unsupervised method for consistent keypoint detection.  "Unsupervised" means the method can be applied in larger-scale datasets without 3D annotations.
2 The paper is easy to follow. The writing is good and clear.
3 Extensive experiments show that this method achieves sota performance on many public datasets

**Weaknesses:**

1 The visualized results presented in this paper all feature uniform and complete point clouds. Can this method handle partial point clouds, such as those obtained from back-projecting depth maps? Can it deal with point clouds that contain severe outliers, not just those with some Gaussian noise? This is very important for practical use.

2 In 289, However, if using a SPRIN module [41] which is a SOTA SE(3)-invariant backbone 289 to replace PointNet++ [22] directly, the training process of Key-Grid does not converge. Why? How about using other backbones? like Vector Neurons?

Deng, C., Litany, O., Duan, Y., Poulenard, A., Tagliasacchi, A., & Guibas, L. J. (2021). Vector neurons: A general framework for so (3)-equivariant networks. In Proceedings of the IEEE/CVF International Conference on Computer Vision (pp. 12200-12209).

3 This proposed representation is based on Euclidean distance between the grid point towards each line segment, right? Why not first establish a graph and then aggregate features along the graph? I saw many papers using graph to model deformable objects.

**Questions:**

I have listed some questions in the weakness. The first question is the most important.

In L 14, 'Meanwhile, we incorporate the information from each layer of the encoder into the decoder section', Is this your contribution?

I recommend to move Fig 6 in the appendix to the main text for clarity.

**Limitations:**

The author has discussed potential limitations in the paper

---

> ### Author Rebuttal · Authors · 2024-08-07
>
> Thanks for your great suggestion. In the rebuttal phase, we address the reviewers' concerns about the effectiveness of our method on partial point clouds, such as those obtained from back-projecting depth map sand point clouds containing outliers. We also provide a detailed explanation of the convergence issues with the SE(3)-invariant model and highlight the advantages of using grid heatmaps over graphs.
>
>
>
> $\color{Indigo}Q1$:  The visualized results presented in this paper all feature uniform and complete point clouds. Can this method handle partial point clouds, such as those obtained from back-projecting depth maps? Can it deal with point clouds that contain severe outliers, not just those with some Gaussian noise? This is very important for practical use.
>
> $\color{red}A$:  In Figure 1(a) and (b), we present the visualization results of keypoints detected by Key-Grid on both point clouds with outlines and partial point clouds sampling from the depth map. We observe that Key-Grid identifies keypoints with strong semantic consistency, even when dealing with point clouds containing ten outliers in Figure 1(a). For the point clouds obtained from depth maps, we first generate depth maps of objects from multiple-angle photographs and then sample point clouds based on the depth maps. We find that  when facing the low-quality/partial point clouds, Key-Grid identifies keypoints whose locations are similar to those detected in the complete point clouds.
>
>
>
> $\color{Indigo}Q2$:  In 289, However, if using a SPRIN module which is a SOTA SE(3)-invariant backbone to replace PointNet++ [22] directly, the training process of Key-Grid does not converge. Why? How about using other backbones? Like Vector Neurons?
>
> $\color{red}A$:  Figure 3 in our new submission material provides the training loss of Key-Grid using SPRIN and Vector Neurons instead of PointNet++ to detect keypoints on the fold pants. By examining the curves, we observe that Key-Grid does not converge by using SPRIN and Vector Neurons in the training phase. We think the SE(3) backbone model is not converging, primarily because of the stringent training conditions. Our designed loss function fails to provide sufficient information to enable the SE(3) model to converge. Like USEEK, to achieve the convergence of the SE(3) model, the authors compute the difference between features outputted by SPRIN and features from the pre-trained PointNet++  in SM as a loss function to optimize the SE(3) model.  If the SE(3) model is directly used as the backbone in SM and Key-Grid, we only optimize it by locating the keypoints instead of providing a standard feature for the SE(3) model to learn from. Therefore, we think this is the main reason why the SE(3) model fails to converge.
>
> $\color{Indigo}Q3$:  This proposed representation is based on Euclidean distance between the grid point towards each line segment, right? Why not first establish a graph and then aggregate features along the graph? I saw many papers using graph to model deformable objects.
>
> $\color{red}A$: Thank you for your valuable perspective. Currently, applying graph networks to deformable objects typically involves constructing a graph structure from point clouds, where nodes represent points on the object and edges capture mesh information between these points. Subsequently, graph neural networks input graph structures and output the required points [1]. However, the role of the grid heatmap is to use keypoints to build the skeletal structure of the deformable object, aiding the network in reconstructing the original point cloud. This process is contrary to the current application of graph neural networks for deformable objects. Therefore, we think that replacing grid heatmaps with graph structures is not suitable for our approach.
>
> [1] Learning language-conditioned deformable object manipulation with graph dynamics. arXiv:2303.01310, 2023.

---

> > ### Comment · Reviewer_LZF7 · 2024-08-10
> > **Final Rating**
> >
> > Thank you for the response. The rebuttal is very good and has addressed my major concerns, and I raise my rating to "weak accept."

---

> ### Author Response · Authors · 2024-08-12
>
> Thank you for your feedback on our paper, particularly regarding the applicability of the Key-Grid to partial and noisy point clouds. We will include this section in the next version of the paper. We greatly appreciate your consideration of the paper as ready for "weak accept".  We notice that the rating is not updated. We kindly hope that you update the rating of our paper to reflect the improvements made to the paper.

---

### Official Review · Reviewer_p6XE · 2024-07-09

**Soundness:** 2
**Presentation:** 3
**Contribution:** 3
**Rating:** 6
**Confidence:** 4

**Summary:**

This paper presents Key-Grid, an unsupervised keypoint detection network designed for 3D point clouds. Unlike previous methods that emphasize leveraging various priors on 3D structures, this paper converts keypoints into a grid heatmap. This heatmap forms a continuous feature landscape across the entire 3D space, providing richer and more stable geometric descriptions of objects, particularly for deformable objects. Meanwhile, this paper achieves state-of-the-art (SOTA) results on the ShapeNetCoreV2 and ClothesNet datasets.

**Strengths:**

1. The proposed grid heatmap is effective in the task of 3D keypoint detection and represents a novel approach to introducing 3D priors for deformable objects.

2. The authors conducted extensive experiments, achieving SOTA performance on both rigid and deformable objects.

3. The overall design follows the mainstream architecture of 3D keypoint detection, providing convenience for subsequent research in this field.

**Weaknesses:**

Q1. My main concern is whether it can be empirically or theoretically justified why the grid heatmap can represent a dense extension of the skeleton, and why the current method of generating the grid heatmap is reasonable. Could using the density of the 'skeleton' or the density of the point cloud directly represent the grid heatmap instead?

Q2.  In the ablation study (Table 4), the accuracy drop after "No Grid Heatmap" is minimal, yet the Grid Heatmap is the most significant contribution of this paper. Could the authors explain this phenomenon and provide ablation studies across more categories?

Q3. Can this paper adaptively select the number of detected keypoints for different categories of objects?

Q4. The quality of figures 1 and 2 could be enhanced. For example, the information content of figure 1 is too minimal to serve as a teaser.

**Questions:**

If the authors address the issues mentioned in the weaknesses, I am willing to increase the rating.

**Limitations:**

Yes, the authors discussed the limitations and potential negative societal impact.

---

> ### Author Rebuttal · Authors · 2024-08-07
>
> Thanks for your insightful comments. We address your concerns as follows: We explain the rationale of the grid heatmap from a theoretical perspective and highlight its advantages over the skeleton structure proposed by SM. Subsequently, we demonstrate  that the grid heatmap is more effective for keypoint detection than skeleton structure on deformable objects through experimental validation.  Furthermore, we clarify some confusions, such as the importance of the grid heatmap to the Key-Grid and whether Key-Grid can adaptively generate keypoints.
>
> $\color{Indigo}Q1$：My main concern is whether it can be empirically or theoretically justified why the grid heatmap can represent a dense extension of the skeleton, and why the current method of generating the grid heatmap is reasonable. Could using the density of the 'skeleton' or the density of the point cloud directly represent the grid heatmap instead?
>
> $\color{red}A$: Skeleton structures proposed by SM does not accurately represent the skeletal structure of deformed objects. For example, in Figure 2 of our paper, if you connect the red keypoint with the black keypoint,  this skeleton does not rationally represent the structure of the folded pants. Unlike SM, which directly uses the lines connecting keypoints as the object's skeleton, we characterize the overall skeletal structure of deformed objects more delicately by depicting the distances between grid points and keypoints pair line. Figure 4(C) of our paper shows the visualization results of the grid heatmaps and skeleton structures proposed by SM， denoting that the grid heatmap provides a more accurate depiction of the underlying skeletal structure of deformed objects. We also report the performance of Key-Grid which replaces the grid heatmap with the skeleton on the fold deformation in the following table. We can conclude that the grid heatmap helps Key-Grid identify keypoints with more semantic relevance on the fold deformation of clothes.
>
> |              | Folded Shirt | Folded Pant |
> |--------------|----------------|--------------|
> | Grid Heatmap |     92.0       |     100.0      |
> | Skeleton  Structure   |     83.9       |     92.7     |
>
>
> $\color{Indigo}Q2$:  In the ablation study (Table 4), the accuracy drop after "No Grid Heatmap" is minimal, yet the Grid Heatmap is the most significant contribution of this paper. Could the authors explain this phenomenon and provide ablation studies across more categories?
>
>
> $\color{red}A$: Key-Grid utilizes the farthest point keypoint loss to pre-select several initial positions for keypoints. Subsequently, we employ the Grid Heatmap and leverage encoder information for hierarchical decoding to assist the model in finalizing these pre-selected positions. In other words, the Grid Heatmap fundamentally aids the model in selecting keypoint positions rather than directly generate their positions. Therefore, when removing the grid heatmap, its overall impact on the model's effectiveness is less significant compared to removing the farthest point keypoint loss, which provides initial candidate keypoint positions.  However, in Table 4 from our paper,  “No Grid Heatmap”   actually has a greater impact on reducing the semantic relevance of keypoints compared to  “No Encoder Information”, which also illustrates the significant role of grid heatmap in the decoder. In the following table, we present an ablation study of more diverse categories of deformable and rigid objects under the DAS metric and also show the average impact of ablating different modules on the performance of Key-Grid.
>
>
> |                         | Table (Rigid) | Motorbike (Rigid) | Hat (Drop) | Long pant (Drop) | Hat (Drag) | Long pant (Drag) | Average  |
> |-------------------------|---------------|-------------------|------------|------------------|------------|------------------|--------------|
> | Key-Grid                    | 78.6          | 63.7              | 100.0        | 83.7             | 51.5       | 99.0             | 79.4          |
> | No Grid Heatmap         | 63.8          | 51.3              | 87.9       | 70.6             | 36.7       | 86.1             | 66.1(-13,3)          |
> | No Encoder Information  | 65.3          | 51.8              | 89.1       | 74.9             | 39.8       | 88.9             | 68.3(-11.1)           |
>
>
> $\color{Indigo}Q3$:  Can this paper adaptively select the number of detected keypoints for different categories of objects?
>
> $\color{red}A$: Thank you for your suggestion. Currently, in Key-Grid, we need to manually set the total number of predicted keypoints beforehand. In  future research, we will propose an adaptive keypoint detection method that can generate varying numbers of keypoints with accurate positions for different samples.
>
> $\color{Indigo}Q4$:  The quality of figures 1 and 2 could be enhanced. For example, the information content of figure 1 is too minimal to serve as a teaser.
>
> $\color{red}A$: Thanks for your advice. In the come ready version, we will submit the new version of Figure 1 and Figure 2. The new version of Figure 1 not only provides the visualizations of keypoints detected on both deformable and rigid objects but also adds the grid heatmap and skeleton structure visualizations like Figure 4(c) in our article, which demonstrates that grid heatmap can more accurately represent the geometric information of the object.

---

> > ### Comment · Reviewer_p6XE · 2024-08-09
> > **Additional comments**
> >
> > Thank you for the response and the additional experiments. My main concerns regarding the grid heatmap representation and ablation study have been resolved. The quality of the Figures 1 and 2 has also improved. I tend to accept this paper. However, I still have another question: can the distances between key points or the density of key points express the dense extension of the deformable object?

---

> > > ### Author Response · Authors · 2024-08-09
> > > **Thanks for your respone**
> > >
> > > In my view, using the distance from a point to keypoints to represent the deformable object’s structure is less effective than using the distance from a point to the skeleton. For example, in the main paper of Figure 2, if we use the distance from a point to keypoints, in the tail section of the pants (i.e., the area between the red and blue points), the value near the red point differs from the value at the center of this section. However, the geometric information represented by these two points should be consistent.
> > > By using the distance from a point to the skeleton, the point values are derived based on the skeleton spreading outward from the center. This results in different values for points near the skeletons compared to those at the object’s edges, reflecting their distinct geometric information. Therefore, we believe that using the distance from a point to the skeleton more accurately characterizes the structure of deformable objects than using the distance to keypoints.

---

> > > > ### Comment · Reviewer_p6XE · 2024-08-10
> > > > **Final Rating**
> > > >
> > > > Thank you for the response. The rebuttal has addressed my main concerns, and I am willing to raise my rating to "weakly accept."

---

### Official Review · Reviewer_U8aQ · 2024-07-10

**Soundness:** 3
**Presentation:** 4
**Contribution:** 3
**Rating:** 7
**Confidence:** 3

**Summary:**

The authors propose a novel unsupervised method to detect 3D point clouds key points by producing an intermediary heatmap based on a grid and points distances from the skeleton and connected key points. They achieve state-of-the-art accuracy and semantic consistency and easily achieve Se-(3) invariance with minimal adaptation of the method.

**Strengths:**

The method's description, results, and a very in-depth ablation study, together with many supplementary materials, make this paper pleasant to read and easy to understand, with impressive results. Although simple at first glance, the method has the merit of being solid and achieving good results. Great results section and analysis allow the reader to understand in depth the method and results.

**Weaknesses:**

A small neat-picking addition would be to add one or two visual examples where the method does not work super well, to contrast with the excellent results presented in the paper

**Questions:**

Perhaps I missed this point, but how does the author ensure that the key points are semantically the same across different folding/deformation of the objects? ( and give them the same colour)

**Limitations:**

Perhaps the only limitation I can pick up is the consistency across more than two deformations of an object. Would the method perform well over long-term deformations? It ties with showing one or two failure cases of the method.

---

> ### Author Rebuttal · Authors · 2024-08-07
>
> Thank you for your valuable feedback. We address your concerns as follows: we provide demonstrations of our method on some less successful objects and investigate whether keypoints maintain semantic consistency across various deformations of objects.
>
> $\color{Indigo} Q1$：A small neat-picking addition would be to add one or two visual examples where the method does not work super well, to contrast with the excellent results presented in the paper
>
> $\color{red}A$:   We present the visual results of the drag deformation for 'Mask' and 'Skirt' in  Figure 4 of our submission material, which exhibit lower DAS values than other categories on the ClothesNet dataset. Although the semantic consistency of keypoints detected on Masks and Skirts is relatively lower than other categories, keypoints identified by Key-Grid uniformly distribute across the object's surface, located at positions rich in geometric information compared to other methods. For instance, keypoints detected by Key-Grid are evenly distributed along the straps of the mask.
>
>
> $\color{Indigo} Q2$：Perhaps I missed this point, but how does the author ensure that the key points are semantically the same across different folding/deformation of the objects?
>
> $\color{red}A$:  Figure 2 of our submission material shows  the visualization results of keypoints detected by Key-Grid on long pants under the dropping, pulling, and dragging deformation. We can observe that keypoints detected by Key-Grid keep great semantic consistency during this long-term deformation process. At the same time, we observe that keypoints identified by Key-Grid are uniformly distributed across deformable objects, with their positions containing substantial geometric information, such as the hem and waist of long pants.

---

### Official Review · Reviewer_YtUf · 2024-07-13

**Soundness:** 3
**Presentation:** 2
**Contribution:** 3
**Rating:** 6
**Confidence:** 4

**Summary:**

Novel PointNet-based autoencoder method  called KeyGrid, that predicts semantic keypoints on objects, even when objects are subject to deformations. Similar to previous approaches, keypoints are produced as a linear combination of inputs points, according to a learned weight matrix.

The key novelty over related works using PointNet++-based auto-encoders such as SkeletonMerger (SM) [26] is to pass a densely sampled grid of features (3d “point heatmap”) at each decoder layer (which are hierarchical). In this heatmap grid, each cell holds the maximum of the weighted distances to the 'keypoint skeleton segments'.
Also, KeyGrid appears to pass richer encoder information to the decoder than the SM work, although the differences here are less clear.

The method is shown to outperform previous baselines such as SK3D and SM on ShapeNetV2 dataset (more right objects) and on ClothesNet (deformable objects).

**Strengths:**

- Beats KeypointDeformer (KD), SkeletonMerger (SM), SC3K on almost all categories on the ClothesNet dataset and ShapeNetCoreV2. Deformable objects keypoints seem to improve more.
- Ablations in Tab. 4 indicate the importance on 4 (among dozens) of semantic classes how encoder-info, a grid heatmap, farthest point sampling loss, and similarity (i.e. chamfer loss) each boost performance.

**Weaknesses:**

# Significance
- Is the hierarchical decoder setup (excluding the heatmap grid itself) novel, esp compared to SkeletonMerger [26]? Right now there are no specific claims or ablations related to this.
- The primary novelty relative to SM [26] seems to be the use of keypoint grid heatmaps in the decoder, storing max distances from grid cells to the weighted keypoint segments. However, this is just one possible shape descriptor feature, and intuitively it should be quite a weak and ad hoc feature, even if the authors gave some intuition why it may be better than the min function. Would other shape feature with yet more information be more useful could (e.g. why not have K channels with distances to all segments or to the keypoints as opposed to just taking the max, etc)? In general it would help if the paper provided more intuition or experimental results giving more insight on the information contained in this heatmap.
- This class of approaches seems generally limited to cases where entire point clouds were visible. How would you handle cases when some parts of the objects were occluded, such as in real-world scenes? This has not beed discussed.

# Related work
Could discuss other dense representations for correspondence like the pointmaps used in “DUSt3R: Geometric 3D Vision Made Easy” https://arxiv.org/pdf/2312.14132 )

# Clarity
- I found Sec 3.3 particularly difficult to follow and understand. A rewrite, or further details in the appendix clarifying better how these quantities are 'composed' and aligned would help.
- Fig 1 illustrates points heatmap in the decoder by simply showing a point cloud. I was expecting pictures like in Fig 4c.
- L13, L58 Unclear where keypoint pairs come from, when discussed for first time.
- No discussion of how correspondences are established, e.g. as shown in Figure 3.
- The loss terms in Equation 11 are not mathematically defined, would be helpful in the Appendix.

# Experimental results
  - KeypointDeformer, SkeletonMerger, and SC3K are tested on KeypointNet dataset (https://arxiv.org/pdf/2002.12687 ). Why do the authors omit it here? Other works such as SC3K state “We use KeypointNet dataset [36 ] in our experiments, considering that this is the standard and most recent dataset used for keypoints estimation”. If the results are not as good on that dataset, that is okay, as long as we provide them and state that our method is specifically designed for datasets with deformable/soft objects. This would provide confidence that the authors ran their eval of official source code correctly on the two new datasets.
  - Why no comparisons with supervised baselines?
  - Figure 4D: Sun3D keypoints are so small in the figure it appears as if they are randomly distributed

# Some Grammar & Style Nits:
- L453 SC3K bibtex should be ICCV 2023, not Arxiv “
- L148 typo “giving a total of 4096 gird points.” -> “…4096 grid points”
- L229 “mean Intersection over Unions” -> “mean Intersection over Union”
- Grammar: Figure 3: caption: “Different methods on the Hat and Long Pant during” ->  prefer “Different methods on the Hat and Long Pant categories during”
- Table 3 caption is too close to Table 3, whitespace seems too shrunken
- L75 “Deformable object dataset” -> “Deformable object datasets”
- Table 1 caption “codes” -> should be singular, not plural (“the results we reproduced based on their official codes” -> “the results we reproduced based on their official code.”)
- Color scheme is jarring in Tables 1-4, prefer tango colors that have a more attractive colormap (https://sobac.com/sobac/tangocolors.htm)
- L3 extraneous article: “focus on the rigid body objects” -> “focus on rigid body objects”
- L6 extraneous article: “for both the rigid-body and deformable objects” -> “for both rigid body and deformable objects”
- L300 “the importance of encoder information and grid heatmap for reconstruction process,” -> “...for the reconstruction process,”
- L302: “Table 4 show that the keypoints detected by Key-Grid which utilizes both two strategies to reconstruct the point cloud have better semantic consistency.” -> “Table 4 shows that when using both input streams to reconstruct the point cloud, the keypoints detected by Key-Grid exhibit better semantic consistency”
- L9-10 Instead of “Unlike previous work, we leverage the identified keypoint information to form a 3D grid feature heatmap 1called grid heatmap, which is used in the decoder section”, prefer to say something like “..to form a 3D grid feature heatmap which we refer to as a grid heatmap. A grid heatmap is…”
- L14 “Into the decoder section” -> “into the decoder model”? Ambiguous what “section” refers to
- L51 “aiming at the semantic consistency” -> “aiming for semantic consistency”, e.g. “an unsupervised keypoint detector on 3D point clouds aiming for semantic consistency under shape variations of both rigid-body and deformable objects.”
- L74 section should be named “Related Work” not “Related Works”

**Questions:**

- Compared to SM[26], is the keypoint heatmap the main difference, how about the use of hierarchical decoder.
- Wouldn't a richer shape descriptor heatmap do even better than a max function? (e.g. K channels, distances to the keypoints, etc).
- How do you get keypoint correspondences across shapes for your model, do they emerge naturally?
- What are the method results on KeypointNet dataset? Can you provide some additional comparisons to supervised baselines on your datasets, if there are such?
- Can this method be applied on datasets where shapes are significantly occluded (e.g. viewed from a specific direction)?

**Limitations:**

Limitations have been sufficiently addressed.

---

> ### Author Rebuttal · Authors · 2024-08-07
>
> Thank you for providing thoughtful and detailed feedback, which greatly enhances the quality of our article.  Based on your comments regarding related work, clarity, and grammar & style nits, we will revise our paper in the came ready section.
> Regarding your inquiries about experimental results and significance sections, we address each one individually. Additionally, we provide the new visualization results of keypoint detected by Key-Grid on the SUN3D dataset in the new submission material.
>
> $\color{Indigo}Q1$:  Compared to SM, is the keypoint heatmap the main difference, how about the use of hierarchical decoder.
>
> $\color{red}A$: The main distinction between SM and Key-Grid lies in the decoder design. We incorporate information from the encoder into the decoder to reconstruct point clouds through hierarchical decoding. Additionally, we integrate grid heatmap information during the reconstruction process. Therefore, we consider the hierarchical decoder and the grid heatmap as the two significant innovations of our method. In our paper, we conduct ablation experiments on the hierarchical decoder in Section 4.5. In Table 4 from our paper, the label “No Encoder Information” means the ablation of the hierarchical decoder.  We observe that when we do not use a hierarchical decoder, our method's performance will decrease on both the deformable and rigid objects.
>
>
> $\color{Indigo}Q2$:  Wouldn't a richer shape descriptor heatmap do even better than a max function? (e.g. K channels, distances to the keypoints, etc).
>
> $\color{red}A$: In the following table, we display the impact of grid heatmaps with different distance definitions on keypoint recognition in deformable objects. These definitions are the K-channel distances from each grid point to all segments or keypoints. For the folding deformation, the distance defined in our paper achieves great performance on the semantic consistency of keypoints. We think that adopting a K-channel distance will result in some overlap of distance information between the point clouds outside the deformable objects and those inside. This condition reduces the ability of the grid heatmap to depict the structure of the deformable objects.
>
> |                                      | Folded Shirt | Folded Pant |
> |--------------------------------------|----------------|--------------|
> | Distance to keypoints     | 86.4           | 93.1         |
> | Distance to segment       |   88.7         | 95.8       |
> | Our                                | 92             | 100          |
>
> $\color{Indigo}Q3$:  How do you get keypoint correspondences across shapes for your model, do they emerge naturally?
>
> $\color{red}A$:  The positions of keypoints are outputted in an orderly manner by Key-Grid and other baselines(SC3K, SM, and KD).  In our article's visualization of keypoints, we use different colors to represent keypoints outputted in different orders. During the entire deformation process, if keypoints of the same color (with the same output order) maintain their positions unchanged, we refer to these keypoints as having good semantic consistency.  The output of keypoints with good semantic consistency can trace various positions that contain essential geometric information in objects when objects undergo deformation. This is crucial for robot manipulation of deformable objects and understanding the deformation process of objects.  In our paper, Key-Grid naturally outputs keypoints with semantic consistency.
>
> $\color{Indigo}Q4$:  What are the method results on KeypointNet dataset? Can you provide some additional comparisons to supervised baselines on your datasets, if there are such?
>
> $\color{red}A$: Thank you for pointing out this issue. In the following table, we present various supervised networks and self-supervised methods to compare Key-Grid on the KeypointNet dataset based on mIoU metric. Several standard networks, PointNet [1], SpiderCNN [2], and PointConv [3], are trained on KeypointNet to predict the probability of each point being a keypoint in a supervised manner. We can observe that compared with other self-supervised methods, Key-Grid recognizes the more accurate location of keypoint, and it performs even better than some supervised methods using PointNet and SpiderCNN as backbones.
>
> |           | Airplane | Chair | Car   | Average |
> |-----------|----------|-------|-------|---------|
> | PointNet  | 45.4     | 23.8  | 15.3  | 28.2    |
> | SpiderCNN | 55.0     | 49.0  | 38.7  | 47.6    |
> | PointConv | 93.5     | 86.0  | 83.6  | 87.0    |
> | SM        | 79.4     | 68.4  | 63.2  | 70.3    |
> | SC3K      | 82.7     | 38.5  | 34.9  | 52.0    |
> | Key-Grid      | 80.9     | 75.2  | 69.3  | 75.1    |
>
> $\color{Indigo}Q5$: Can this method be applied on datasets where shapes are significantly occluded (e.g. viewed from a specific direction)?
>
> $\color{red}A$:  Thanks for your insightful suggestion. Keypoints detected by Key-Grid estimated on the deformation of objects from side views are depicted in Figure 1 from the new submission material. Meanwhile, we also show the keypoint detection results when objects' shapes are occluded in Figure 1 from the new submission material. We find that Key-Grid exhibits robustness on the occluded point clouds and point clouds obtained from different views.
>
> [1] Pointnet: Deep learning on point sets for 3d classification and segmentation. CVPR 2017
>
> [2] Spidercnn: Deep learning on point sets with parameterize convolutional filters. ECCV 2018
>
> [3] Pointconv: Deep convolutional networks on 3d point clouds. CVPR 2019

---

> > ### Comment · Reviewer_YtUf · 2024-08-10
> > **rebuttal response**
> >
> > The authors have provided helpful answers and results that largely address my questions. I also saw the newly attached figure, which shows performance under some occlusion - which is ok even if the occlusion amount shown is quite small.
> >
> > In light of this, I am willing to raise my rating to 'weakly accept'.

---

> ### Author Response · Authors · 2024-08-12
>
> Thank you for your detailed review and comments on our paper. We will revise our paper according to your  suggestions. We are also pleased to hear that you believe our paper has reached the ’weak accept‘ stage.  However, we observe that the rating is not updated. We would be grateful if you could revise the rating to better reflect the improvements made to the paper.

---

### Author Rebuttal · Authors · 2024-08-07

Thanks to the esteemed reviewers for your insightful feedback, which has significantly enhanced the quality of our paper. Based on your suggestions, we provide the corresponding visual results in the new submission material.

**Figure 1(a)**: In response to Reviewer YtUf and Reviewer LZF7's inquiries regarding our method's capability to handle the occluded, partial, and outlier-laden point clouds, we show the visualization results of Key-Grid's performance on these types of point clouds.

**Figure 1(b)** : Regarding the Reviewer LZF7's inquiry about whether Key-Grid can handle point clouds obtained from depth maps, we generate a depth map from multi-angle images and then illustrate keypoints identified by Key-Grid on the point cloud sampled from this long pant depth map.

**Figure 2**: For Reviewer U8aQ's concern about the effectiveness of Key-Grid on objects undergoing  long-term deformation, we demonstrate the KeyGrid's capability to capture the keypoints with high semantic consistency during the long-term deformation processes containing dropping, pulling, and dragging. In response to the YtUf reviewer's concern about the visualization result of keypoints in the Sun3D dataset is not clear, we improve the visibility of the keypoints and present additional visualization results with more samples from the Sun3D dataset.

**Figure 3**: Regarding the LZF7 reviewer's concern about the non-convergence of Key-Grid with the SE(3)-invariant backbone, we present the training loss curves of the Key-Grid method with the PointNet+/SPRIN/Vector Neurons models.

**Figure 4**: In response to the U8aQ reviewer's request to show some examples where our method does not work well, we select the visualizations of keypoint detected on the Skirt and Mask under the drag deformations.

---

### Decision · Program_Chairs · 2024-09-25

**Decision:**

Accept (poster)

**Comment:**

The paper proposes a novel approach for 3d keypoint estimation via unsupervised learning. It initially received discordant ratings and comments from the reviewers (BR, BR, BA, SA). During the rebuttal and discussion stage, some main concerns from the reviewers have been resolved and a stronger consensus towards acceptance emerged. 3 reviewers suggested to accept (6,6,7) and one reviewer kept the initial BR score but mentioned in their comments that they are satisfied with the rebuttal and open to raise their score. Overall, the AC suggests to accept this paper as their final recommendation.